# Assessment of Gastrointestinal Symptoms and Dyspnea in Patients Hospitalized due to COVID-19: Contribution to Clinical Course and Mortality

**DOI:** 10.3390/jcm11071821

**Published:** 2022-03-25

**Authors:** Krzysztof Kaliszewski, Dorota Diakowska, Łukasz Nowak, Urszula Tokarczyk, Maciej Sroczyński, Monika Sępek, Agata Dudek, Karolina Sutkowska-Stępień, Katarzyna Kiliś-Pstrusińska, Agnieszka Matera-Witkiewicz, Michał Pomorski, Marcin Protasiewicz, Janusz Sokołowski, Barbara Adamik, Krzysztof Kujawa, Adrian Doroszko, Katarzyna Madziarska, Ewa Anita Jankowska

**Affiliations:** 1Department of General, Minimally Invasive and Endocrine Surgery, Wroclaw Medical University, Borowska Street 213, 50-556 Wroclaw, Poland; urszula.tokarczyk@student.umw.edu.pl (U.T.); maciej.sroczynski@o2.pl (M.S.); moniasep@wp.pl (M.S.); agatadudek7@gmail.com (A.D.); karolina.sutkowska@onet.pl (K.S.-S.); 2Department of Basic Science, Faculty of Health Science, Wroclaw Medical University, Bartel Street 5, 51-618 Wroclaw, Poland; dorota.diakowska@umed.wroc.pl; 3Department of Urology and Urological Oncology, Wroclaw Medical University, Borowska Street 213, 50-556 Wroclaw, Poland; lllukasz.nowak@gmail.com; 4Department of Pediatric Nephrology, Wroclaw Medical University, Borowska Street 213, 50-556 Wroclaw, Poland; katarzyna.kilis-pstrusinska@umed.wroc.pl; 5Screening Laboratory of Biological Activity Tests and Collection of Biological Material, Faculty of Pharmacy, Wroclaw Medical University, Borowska Street 211A, 50-556 Wroclaw, Poland; agnieszka.matera-witkiewicz@umed.wroc.pl; 6Department of Gynecology and Obstetrics, Wroclaw Medical University, Borowska Street 213, 50-556 Wroclaw, Poland; michal.pomorski@umed.wroc.pl; 7Department and Clinic of Cardiology, Wroclaw Medical University, Borowska Street 213, 50-556 Wroclaw, Poland; marcin.protasiewicz@umed.wroc.pl; 8Department of Emergency Medicine, Wroclaw Medical University, Borowska Street 213, 50-556 Wroclaw, Poland; janusz.sokolowski@umed.wroc.pl; 9Department of Anaesthesiology and Intensive Therapy, Wroclaw Medical University, Borowska Street 213, 50-556 Wroclaw, Poland; barbara.adamik@umed.wroc.pl; 10Statistical Analysis Center, Wroclaw Medical University, Marcinkowski Street 2-6, 50-368 Wroclaw, Poland; krzysztof.kujawa@umed.wroc.pl; 11Department of Internal Medicine, Hypertension and Clinical Oncology, Wroclaw Medical University, Borowska 213, 50-556 Wroclaw, Poland; adrian.doroszko@umed.wroc.pl; 12Department of Nephrology and Transplantation Medicine, Wroclaw Medical University, Borowska Street 213, 50-556 Wroclaw, Poland; katarzyna.madziarska@umed.wroc.pl; 13Centre for Heart Diseases, Department of Heart Diseases, Wroclaw Medical University, Borowska Street 213, 50-556 Wroclaw, Poland; ewa.jankowska@umed.wroc.pl

**Keywords:** COVID-19, SARS-CoV-2, gastrointestinal manifestations, abdominal pain, dyspnea, clinical course, mortality

## Abstract

Gastrointestinal manifestations may accompany the respiratory symptoms of COVID-19. Abdominal pain (AP) without nausea and vomiting is one of the most common. To date, its role and prognostic value in patients with COVID-19 is still debated. Therefore, we performed a retrospective analysis of 2184 individuals admitted to hospital due to COVID-19. We divided the patients into four groups according to presented symptoms: dyspnea, *n* = 871 (39.9%); AP, *n* = 97 (4.4%); AP with dyspnea together, *n* = 50 (2.3%); and patients without dyspnea and AP, *n* = 1166 (53.4%). The patients with AP showed tendency to be younger than these with dyspnea, but without AP (63.0 [38.0–70.0] vs. 65.0 [52.0–74.0] years, *p* = 0.061), and they were more often females as compared to patients with dyspnea (57.7% vs. 44.6%, *p* = 0.013, for females). Patients with AP as a separate sign of COVID-19 significantly less often developed pneumonia as compared to individuals with dyspnea or with dyspnea and AP together (*p* < 0.0001). Patients with AP or AP with dyspnea were significantly less frequently intubated or transferred to the intensive care unit (*p* = 0.003 and *p* = 0.031, respectively). Individuals with AP alone or with dyspnea had significantly lower rate of mortality as compared to patients with dyspnea (*p* = 0.003). AP as a separate symptom and also as a coexisting sign with dyspnea does not predispose the patients with COVID-19 to the worse clinical course and higher mortality.

## 1. Introduction

In December 2019, a large number of pneumonia cases of unknown origin were first reported in Wuhan, Hubei Province, China [1]. Next, after several weeks, the new pathological agent was recognized as a novel coronavirus responsible for this severe acute respiratory syndrome (SARS). The virus was named SARS coronavirus 2 (SARS-CoV-2) as a member of the Coronaviridae family [1,2]. This group of viruses is routinely presented among animals and humans, and contains nonsegmented enveloped RNA viruses with a single-strand linear positive-sense RNA (2). Subsequently, the disease caused by SARS-CoV-2 was introduced as coronavirus disease-2019 (COVID-19) [1,2,3]. The new, unknown virus disease spread rapidly worldwide, and 11 March 2020 was established as the start date of the global pandemic by the World Health Organization (WHO) [3]. Since that time, over 212 million patients with a COVID-19 diagnosis and 4.4 million COVID-19-associated deaths have been observed [4]. The prevalence of COVID-19 is steadily increasing [5].

Initially, according to many studies, COVID-19 was recognized only as a respiratory tract infection. However, later, many authors described COVID-19 as a disease with a wide range spectrum of symptoms due to its involvement of not only the lungs but also other organs [6,7]. Hoffmann et al. [8] revealed that it is caused by the interaction of SARS-CoV-2 with other human organs mediated by angiotensin converting enzyme 2 (ACE 2). ACE 2 is expressed on many anatomical structures on the cell surface. ACE 2 was identified as an integral membrane protein that is recognized as the host cell receptor for SARS-CoV-2. Additionally, Chen et al. [9] observed ACE 2 as the host cell receptor in a large quantity in patients with COVID-19. Varga et al. [10] established that endothelial ACE 2-positive cells obtained from COVID-19 patients present significant morphological changes, such as disruptions of intercellular junctions, cell swelling, and damaged contacts of the basement membrane. Despite the fact that ACE 2 is mostly localized in the alveolar epithelium, some authors have highlighted its presence in the liver and biliary epithelium, enterocytes, or the vascular endothelium [11,12]. Therefore, it can be expected that such pathophysiological reactions may lead to some other clinical manifestations of the extrapulmonary locations.

It was estimated that on 19 January 2020, in Washington (USA), the first COVID-19-positive patient manifested, in addition to respiratory sings, abdominal symptoms such as abdominal pain (AP), nausea, and vomiting [13]. Some authors noticed that several individuals with COVID-19, who presented to the emergency department (ED) had only AP without any typical respiratory signs characteristic of this infection [1]. Others described an interesting situation, that was individually observed in EDs: in patients with severe AP, when radiologists subsequently performed abdominal computed tomography [CT] due to the abdominal symptoms, they recognized the SARS-CoV-2 infection due to the typical findings of peripheral and subpleural ground-glass opacities in the lower lobes of the lung [14,15,16]. Regarding some observational studies describing connections between COVID-19 and digestive system failure symptoms [17,18,19], we can ask whether they should be treated as specific signs for SARS-CoV-2 infections, especially in patients without any respiratory symptoms. However, there are also other important issues, which in our opinion stand in contrast to previously mentioned ones. Namely, some authors recognize AP in patients with COVID-19 as a side effect of virus infection treatment and as secondary to systemic inflammation and ischemia [20]. Others say that AP may also occur in patients with basilar pneumonia [12,21]. However, many authors have reported an association between SARS-CoV-2 infections and other abdominal diseases, such as acute pancreatitis, which has been well documented by clinical studies [22,23,24,25,26]. On the other hand, there are analyses that have considered such observations, even well-documented ones, inappropriate [27]. These authors recommend treating the relationship between SARS-CoV-2 infections and, for instance, mentioning acute pancreatitis as a casual association, due to insufficient etiological evaluation.

Due to these observations and many doubts about the COVID-19 disease, we performed a retrospective analysis of 2184 individuals with COVID-19 to determine whether, and how many patients infected by SARS-CoV-2 manifested AP. Second, we checked, whether the patients with COVID-19 disease, who presented with AP, had significantly elevated specific markers for digestive system involvement. Third, we assessed the correlation between respiratory symptoms and AP to establish how often they occur together and separately. Finally, we assessed how AP may influence the clinical course and mortality of patients with COVID-19.

## 2. Materials and Methods

### 2.1. Study Population

We retrospectively analyzed 2184 medical records of patients admitted and treated at the university and temporary COVID-19 hospital arranged by the Medical University Hospital in Wroclaw (Poland) between February 2020 and June 2021.

The study protocol was approved by the Institutional Review Board and Ethics Committee of Wroclaw Medical University, Wroclaw, Poland (No: KB-444/2021). All our patients provided informed consent for admission into the study, which stipulated that the results may be used for research purposes. The data were analyzed retrospectively and anonymously from established medical records. The authors did not have access to identifying patient information or direct access to the study participants.

The patients were divided into four groups according to presented symptoms at the time of admission to the hospital: group A—patients without dyspnea and AP, (study control population); group B—patients with dyspnea; group C—patients with AP; and group D—patients with dyspnea and AP. Patients in Group A did not have dyspnea and AP, and they were included to this group as the patients with COVID-19 positive test and other signs of virus infection (except of AP and dyspnea): cough, fever, pain in the chest, standstill, stressed sounds, and whistling sounds heard over the lung fields, diarrhea, vomiting, deterioration of health, weakness, and elevated inflammatory parameters. The patients infected by SARS-CoV-2 in whom we confirmed acute abdominal disease such as appendicitis, cholecystitis, diverticulitis, incarcerated and strangulated abdominal hernia, occlusion of mesenteric artery or aortic aneurysm were excluded from the study (see Appendix A). They were transported to the department of surgery for potential surgical intervention. They were not admitted to the temporary COVID-19 hospital.

### 2.2. Statistical Analysis

Descriptive data were presented as number of observation and percent (for categorical variables) or as mean, standard deviation (±SD), median and interquartile range (Q25–Q75) (for continuous variables). Distribution of continuous variables was tested by Kolmogorov-Smirnoff with Lillefor’s correction or Shapiro-Wilk normality tests. Chi-square test and Fisher exact test were used for comparison of qualitative variables. Quantitative variables were tested by ANOVA Kruskal-Wallis analysis with post-hoc Dunn test with Bonferroni correction. A significance level of 0.05 was assumed in the analyzes.

Univariable and multivariable logistic regression analysis was used for the determination of independent clinical and laboratory predictors of patients’ death. Results of *p* > 0.1 in univariable logistic analyzes were assumed as exclusion criterion in multivariable analysis. Odds ratios (OR) and confidence intervals (±95% CI) were calculated. In some cases, it was observed a strong correlation between the variables (r = 1.0, *p* < 0.0001), the matrix model was improperly conditioned and the criteria for logistic regression analysis were not met, these variables were removed from the analysis. Clinical and laboratory variables were analyzed separately in multivariable log analysis.

Statistical calculations were performed using Statistica v.13.3 (Tibco Software Inc., Palo Alto, CA, USA).

## 3. Results

### 3.1. Patient Characteristics

The study group consisted of 2184 COVID-19 patients. There were 1102 (50.5%) women and 1082 (49.5%) men, with a mean age of 60.1 ± 18.8 years old. Group A consists of 1166 (53.4%) patients, group B contains 871 (39.9%) individuals, group C 97 (4.4%) patients, and group D consists of 50 (2.3%) patients. Groups were different in sex (*p* = 0.0001) and age (*p* = 0.001) parameters (Table 1), therefore subsequent data were analyzed separately according sex. The “age” variable was not significant in selected “women” and “men” groups in post-hoc tests (*p* > 0.05).

Baseline characteristics of patients with COVID-19 according to clinical symptoms reported by patients during admission time to the hospital was presented in Table 1.

The patients with dyspnea (Group B) were significantly older than patients without respiratory sign and AP (Group A) (65.0 [52.0–74.0] vs. 63.0 [41.0–73.0] years, *p* = 0.002), and they showed tendency to be older than patients with AP (Group C) (*p* = 0.061). The patients with dyspnea were more often male as compared to other study groups (55.4% vs. 46.0% or 42.0%, *p* < 0.0001).

### 3.2. Relationship between Presented Symptoms in Four Study Groups and Other Clinical and Laboratory Parameters Obtained on Admission to Hospital and during Hospitalization

Table 2 showed the prevalence of clinical factors and results of laboratory parameters in COVID-19 patients divided into four study groups in two time periods, namely at the time of admission to the hospital and during hospitalization.

At the time of admission to the hospital we observed statistically significant correlation between the prevalence of the clinical signs characteristic for analyzed groups. In the groups of patients with dyspnea (Group B) and with dyspnea and AP (Group D), the clinical respiratory signs (cough, standstill, stressed sounds, and whistling sounds heard over the lung fields) were observed significantly more often as compared to the patients without dyspnea (Group A) and to the patients presented only AP (Group C) (*p* < 0.0001 for all) (Table 2). Similarly, the clinical signs characteristic for patients with AP (Group C) and AP with dyspnea (Group D) (diarrhea and vomiting) were significantly more often observed in these groups comparing to the groups without AP (Groups A and B) (*p* < 0.0001 for all) (Table 2). Saturation (SaO_2_) without oxygen therapy significantly differed between comparing groups (A vs. B, *p* < 0.0001, A vs. D, *p* = 0.0004; B vs. C, *p* < 0.0001; C vs. D, *p* < 0.001) what correlated with clinical symptoms presented by patients (Table 2). Consequently, respiratory supports (oxygen mustache cannula, face mask, Venturi mask, passive oxygen therapy, High Flow Nasal Cannula (HFNC), Bilevel Positive Airway Pressure/Continuous Positive Airway Pressure (BiPAP/CPAP), and lastly respirator therapy) were significantly more often introduced in patients with dyspnea and with dyspnea and AP (Groups B and D). However, none of the patients with AP and AP with dyspnea did not need HFNC, BiPAP/CPAP, and respirator therapy. What is interesting, we observed significantly higher levels of aspartate transaminaze (ASPAT) and alanine aminotransferaze (ALAT) in the blood of the patients presented dyspnea separately and with AP (Groups B and D) when compared to the patients with AP alone (Group C) (A vs. B, *p* < 0.0001; B vs. C, *p* = 0.009; A vs. B, *p* < 0.0001) (Table 2). We observed the analogous situation with C-reactive protein (CRP) in compared groups (A vs. B, *p* < 0.0001; A vs. D, *p* < 0.001; B vs. C, *p* < 0.001) (Table 2). We did not observe statistically significant correlations in analyzed groups in case of blood levels of total bilirubin, amylase, and procalcitonin.

During hospitalization the patients with dyspnea and with dyspnea and AP (Groups B and D) presented significantly more often deterioration of health what needed maximum aggressive oxygen therapy (*p* < 0.0001) (Table 2). The most aggressive respiratory support during hospitalization (HFNC, BiPAP/CPAP and respirator therapy) was significantly more often introduced in patients with dyspnea (Group B) as compared to the patients with AP or with AP and dyspnea (Groups C and D). Patients with AP (Group C) significantly less frequently developed pneumonia compared to patients with dyspnea and patients with dyspnea and AP (Groups B and D) (*p* < 0.0001) (Table 2). Patients with AP and AP with dyspnea (Groups C and D) were significantly less frequently intubated and transferred to the Intensive Care Unit as compared to the patients with dyspnea (Group B) (*p* = 0.003 and *p* = 0.031, respectively) (Table 2). Patients with AP as a separate sign (Group C) and with AP and dyspnea (Group D) had significantly lower rate of mortality as compared to patients with dyspnea as a separate symptom (Group B) (*p* = 0.003) (Table 2). We observed significantly higher levels of ASPAT and ALAT in the blood of the patients presented dyspnea separately and with AP (Groups B and D) when compared to the patients with AP alone (Group C) (A vs. B, *p* < 0.0001; A vs. B, *p* < 0.0001; B vs. C, *p* = 0.003) (Table 2). During hospitalization we observed significantly higher level of CRP in patients with AP and dyspnea (Group D) (*p* = 0.006) (Table 2).

### 3.3. Selected Determinants of COVID-19 Patient Death

In order to test the risk of qualifying a COVID-19 patient to a given group on the basis of the presented symptoms at the time of admission to the hospital, we conducted a logistic analysis for three study groups (A, B and C) as predictors of death (Table 3).

We analyzed numerous risk factors and potential predictors of patient’s death. Univariable logistic regression analysis was used for precise assessment of each parameter’s influence. Then two multivariable models of selected determinants were obtained for patients with dyspnea (Group B, *n* = 871) and patients with AP (Group C, *n* = 97). A corresponding model for group of patients with dyspnea and AP (Group D, *n* = 50) was not possible due to the low number of subjects.

In the univariable logistic analysis for patients with dyspnea as a separate symptom (Group B), majority of tested risk factors significantly influenced the prevalence of patients’ death (*p* < 0.05 for all) (Table 4).

Model of multivariable analysis presented for patients with dyspnea alone (Group B) (Table 4) revealed, that an increase in age above 65 years old, male gender, standstill heard over the lung fields, passive oxygen therapy, HFNC therapy, and also levels of amylase in blood above 160 U/L (normal range: 20–160 U/L) and procalcitonin above 0.1 ng/mL (normal range <0.1 ng/mL) at the time of admission to the hospital were connected with significantly higher risk of death (*p* < 0.05 for all). Predictors of patients’ death in patients with dyspnea without AP (Group B) during hospitalization were HFNC, BiPAP/CPAP, cardiac and septic shock, the need for intubation, transfer to the intensive care unit, ASPAT above 40 U/L (normal range: 0–40 U/L), total bilirubin above 1.1 mg/dL (normal range: 0.3–1.1 mg/dL), CRP above 5 mg/L (normal range: 3.0–5.0 mg/L), and elevated procalcitonin above 0.1 ng/mL (*p* < 0.05 for all) (Table 4).

In the univariable logistic regression analysis for patients with AP as a separate symptom (Group C), respirator therapy, pneumonia, hypovolemic shock, septic shock, transfer to the intensive care unit, procalcitonin level above 0.1 ng/mL at the time of admission to the hospital, and during hospitalization were predictors of patient’s death (*p* < 0.05). Model of multivariable analysis constructed for patients with AP (Group C) (Table 5) revealed that increase of procalcitonin level above 0.1 ng/mL at the time of admission to the hospital and during hospitalization was the only risk factor of patients’ death (*p* < 0.05) (Table 5).

## 4. Discussion

Since the COVID-19 pandemic is ongoing, it was estimated, that although respiratory signs are predominant, extrapulmonary manifestations are also present. Clinical observations revealed, that almost all organs and systems can be involved in SARS-CoV-2 infections [28]. Guan et al. [15] suggests, that among many others, AP may occur as an isolated clinical finding in patients with COVID-19. Digestive symptoms such as nausea, vomiting, and diarrhea may accompany respiratory symptoms. In our study group, we estimated that patients with dyspnea without AP presented diarrhea and vomiting in 6.7% and 2.4% of cases, respectively. In patients with AP and dyspnea, digestive symptoms were presented more often (i.e., diarrhea in 26.0% of the patients and vomiting in 24.0%). A similar situation was observed in patients presenting only AP without dyspnea. In this group we observed diarrhea in 21.7% and vomiting in 30.9% of the patients.

Sanku et al. [29] noticed, that gastrointestinal bleeding is a rare and unusual finding since patients with COVID-19 are often hypercoagulable, so they are more likely to clot than to bleed. In our study, we noticed digestive tract hemorrhage from the upper part in 1.4% of the patients presenting dyspnea. Interestingly, we did not observe this symptom in patients with dyspnea and AP and only observed digestive tract hemorrhage from the upper part in 3.1% of the patients with AP but without dyspnea. Bleeding from the lower part of the digestive tract was only observed in 0.3% of the patients presenting dyspnea at admission, in 2.0% of the patients with AP, and dyspnea in 1.0% of the patients, who presented only AP as an isolated symptom. We also observed hemorrhage from the respiratory tract, especially in patients, who presented only dyspnea as a main clinical symptom of COVID-19. In this group we observed respiratory tract hemorrhage in 17 (2.0%) individuals.

In addition to respiratory parameters, liver, biliary tract, and pancreatic laboratory tests are routinely ordered in patients with SARS-CoV-2 infections, especially in those presenting with abdominal complaints. In our study we did not perform the analysis of the imaging tests of the abdominal cavity (CT, ultrasonography, endoscopy) due to low number of the completed examinations. Despite good evidence of a correlation between elevated pancreatic enzymes, liver and biliary injury markers, and diseases of these organs, there is still a lack of data regarding these observations during SARS-CoV-2 infections and the course of the disease. Generally, in our analysis we did not observe any significant correlation of elevated pancreatic and hepatic enzymes, and abdominal complaints. In the analyzed groups of patients with dyspnea (Group B), AP (Group C), and dyspnea with AP together (Group D) were investigated for women and men at the time of admission and during hospitalization, and we noticed only two significant differences. Interestingly, in contrast to other studies, we noticed that the blood level of aspartate transaminase (ASPAT) was significantly higher in the groups of patients with dyspnea as a main clinical symptom and with AP (Groups B and D), but not in patients with AP (Group C). This significant difference was also seen during hospitalization. At the time of admission and during hospitalization we observed, that the blood level of alanine aminotransferase (ALAT) was significantly higher in the groups of patients with dyspnea and AP with dyspnea (Group B and D), but not in the group with AP alone (Group C vs. A and B). Revzin et al. [30] estimated that liver failure in patients with COVID-19 is rather mild and transient, but severe liver injury might be diagnosed in the case of sepsis and coagulopathy with microthrombosis. In our study, we confirmed a higher rate of sepsis in the group with dyspnea. Bhayana et al. [31] established that bowel injury in patients with SARS-CoV-2 infection is most likely caused by direct virus spreading into the gut epithelium or by small vessel thrombosis with subsequent ischemia. However, Balaban et al. [12] noticed, that other pathologies are also observed, such as hemorrhagic complications leading to hematomas in the gut walls. Some authors have established potential proposals for the mechanisms of digestive system involvement in patients infected with SARS-CoV-2 [20,32,33]. It might be the explanation for our observations, that in the groups of patients, who complained of AP as a separate symptom or with dyspnea (Groups C and D), vomiting, and diarrhea were observed significantly more often. For hepatobiliary injury, some authors proposed direct viral cytopathic damage, congestive hepatopathy, drug-induced liver injury, systemic inflammatory response and exacerbation of preexisting chronic liver disease. For pancreas failure, some authors proposed direct viral cytopathic injury, systemic inflammation, and dehydration. In the end, gastrointestinal tract damage, according to these authors, is mainly caused by virus cytopathic injury, systemic inflammation, thrombosis, and adverse effects of COVID-19-related drugs [12,20,32,33]. Additionally, some other authors state that the presence of SARS-CoV-2 within endothelial cells, in addition to direct viral effects, also produces perivascular inflammation [10]. Han et al. [17] assessed that 57% of COVID-19-positive patients with a low severity of disease manifested AP alone or in combination with respiratory symptoms. In our study, there was definitely a lower number of patients with AP alone or with dyspnea, at 6.7%. Liu et al. [34] revealed that clinically observed AP in patients with COVID-19 infection usually comes from involvement of the gastrointestinal tract and hepatobiliary-pancreatic system, whereas urinary tract failure and spleen involvement were not commonly observed. In our study, we did not observe any symptoms from a spleen or urinary tract injury, so we did not note any details in our clinical base.

Jutzeler et al. [35] noticed that men are more commonly affected by SARS-CoV-2 infections, and their hospital mortality rate is significantly higher than that of women infected with SARS-CoV-2. In our study we confirmed, that men were significantly more often observed in the group of patients with dyspnea as a separate sign (Group B). However, they more rarely complained of AP than women. The mortality rate was also higher in the group of patients, who complained of dyspnea alone during hospitalization than in the other groups. What is interesting, regarding the patients, who complained of AP (separately or with dyspnea), the mortality rate was at the same level (18.1 vs. 10.3 and 10.0). Han et al. [17] noticed, that patients with abdominal symptoms, compared to those presenting with only respiratory difficulties, had a longer time between the onset of the disease symptoms and viral clearance. They also observed, that patients with digestive components were referred later to medical care units than those with pulmonary symptoms caused by SARS-CoV-2. The delay might be due to these nonspecific manifestations of COVID-19. The authors also identified SARS-CoV-2 in stool samples in a proportion of COVID-19-positive individuals [17]. Hung et al. [36] identified replication of SARS-CoV-2 in the wall of the small and large intestines. Varga et al. [10] analyzed the small intestine of two infected patients and confirmed endotheliitis of the submucosa vessels with mononuclear cell infiltrates within the intima along the lumen of the vessels. They also noticed direct virus presence in endothelial cells. Effenberger et al. [37] confirmed elevated fecal calprotectin concentrations as the result of the inflammatory response in the intestines of patients with SARS-CoV-2 infections, who presented with diarrhea. Elevated concentrations were not observed in the individuals without diarrhea. Goldberg-Stein et al. [38] performed a retrospective analysis of 141 patients with COVID-19 and revealed, that the most common gastrointestinal symptom in these patients was AP. It was observed in 73.8% of the individuals with negative abdominal CT scans and in 53.8% of the patients with positive abdominal CT findings [38]. The authors noticed that 64% of COVID-19-positive patients without any pathological signs in abdominal CT scans had characteristic changes for SARS-CoV-2 infections at the lung bases [38]. Thus, some other authors suggest, that patients with COVID-19 disease without any pathological signs of the abdominal organs may present AP and may have secondary pleural inflammation [39]. Durmus et al. [40] noticed that during the COVID-19 pandemic, it is important to consider a diagnosis of COVID-19 disease in patients with non-severe flank pain if no urological pathology is evident on abdominal CT scans. In our study, a large majority of the patients did not have any imaging tests performed, what we included in the limitations of the study.

The gastrointestinal lesions produced by COVID-19 pose a major challenge in meeting individuals’ nutritional needs. Szefel et al. [41] noticed that the lack of nutrients for the intestinal mucosa may produce atrophy of the lymphoid tissue and subsequently cause immune system deficiency and intestinal bacterial translocation. According to some authors’ observations, diarrhea was the first symptom of COVID-19, before respiratory involvement, and sometimes might even be the only sign of the disease [42,43]. In our study, diarrhea was observed in 5.8% of all patients.

In the end, it is worth emphasizing that the liver was estimated to be the second most injured organ following the lungs in patients infected by the SARS-CoV-2 virus [44]. ACE 2 receptors are widely expressed, especially on cholangiocytes, and even more expressed on cholangiocytes than on hepatocytes [4,45]. Zhao et al. [46] described the virus’s damaging effects on the bile acid barrier in cholangiocytes and the disruption of genes, which destroy cell connections and bile acid transport. According to some authors, the immune-mediated cytokine storm is often included in liver damage [1]. The authors observed elevated plasma levels of C-reactive protein, lymphocytes, neutrophils, and some cytokines, especially interleukin-6. Nardo et al. [47] suggested that the effort to stop cytokine dysregulation at the very beginning of COVID-19 disease may decrease the progression of liver injury. All pathological events with respiratory-induced liver hypoxia and drug-induced liver toxicity produce coagulopathy and consequently, cause damaging changes in microcirculation with microthrombosis within the liver sinusoids [15]. According to Han et al. [17], 14.0% of patients with COVID-19 present elevated serum ASPAT, ALAT, and total bilirubin (TBIL), what is evidence of liver injury. Zhang et al. [6] noticed that 50% of SARS-CoV-2-infected patients presented elevated plasma levels of gamma-glutamyl transferase (GGT). Individuals with elevated liver laboratory tests were at higher risk of severe disease progression, which becomes more noticeable within the first two weeks of hospitalization [4]. ALAT, ASPAT, TBIL, and GGT levels can be elevated to more than three times the upper limit of the normal range [48]. Additionally, some authors state that patients with severe liver injury have a higher rate of intubation and dializotherapy than patients with mild, moderate or no liver injury during the course of COVID-19 disease [1]. According to an analysis presented by the Institute of the American Society of Gastroenterology, more than 60% of patients with COVID-19 have mild liver injury [21]. Of course, the liver condition of patients during SARS-CoV-2 infections depends on the hepatic disease history and the liver function before COVID-19 disease, such as chronic hepatitis or steatosis. Some authors estimated the undisputed impact of pre-existing liver diseases in patients with COVID-19 and its course [49]. They noticed that the risk of hospitalization and death in such individuals was significantly higher.

AP caused by pancreatic involvement in patients with SARS-CoV-2 infections has also been reported [50]. Although, the pathogenetic mechanisms are not as well described for pancreatic involvement as for liver injury, the most likely reasons are the cytopathic effect of the virus and immune-mediated storm. Liu et al. [50] estimated that 17.9% and 16.4% of patients with COVID-19 had elevated plasma amylase and lipase levels, respectively. Others highlighted elevated blood glucose levels as the next marker of pancreatic injury during a COVID-19 infection [51].

In addition to this fact, that we did not observe such lesions, according to some authors, the spleen is the next abdominal organ involved in SARS-CoV-2 infections [52]. In their opinion, ACE 2 receptors are also localized on red pulp and vascular endothelial cell surfaces. Xu et al. [52] stated, that the virus may directly influence macrophages and dendritic cells. These authors reported splenic parenchyma congestion, hemorrhage and lymphatic vesicle absence with spleen parenchyma atrophy, which were noticed during the autopsies of patients, who died of COVID-19. AP originating from splenic injury is caused by splenomegaly and solitary or multifocal splenic infarcts [52].

The outcomes of COVID-19 patients reported by many authors suggest an association between the presence of AP and important clinical observations such as delay in presentation, disease severity and mortality [53]. In summary, we can say that gastrointestinal and respiratory involvement in COVID-19 often occur together. This is probably why Zhou et al. [54] started to promote the concept of the “gut-lung axis”. They suggested that stimulation on one side triggers a response on the other side in patients infected by SARS-CoV-2 [54]. However, in our opinion, there is still too few evidence and completed pathophysiological studies to promote such concept.

Our work has some limitations. First, it was a retrospective study, and access to some necessary specific details was limited. Second, this study was observational, which makes it difficult to control for all potential confounding factors, including age, sex, smoking or vaccination status. Third, there was selection bias since patients included in this study were admitted to the hospital, which indicates that the patients were not representative of the whole population. Fourth, the analyzed data came from a single medical center, so the possibility of selection bias cannot be ruled out. Fifth, we performed our study and made conclusions after an analysis of clinical and biochemical parameters, which only indirectly indicated their pragmatic value in disease prognosis. The large majority of patients did not have imaging tests of the abdominal cavity performed, so we did not decide to include the analysis of them in our study. Many parameters of COVID-19 patients did not show any significant correlation with the clinical course of the disease. This might be since our study included a relatively small number of individuals.

To conclude, AP as a separate symptom and also as a coexisting sign with dyspnea does not predispose the patients with COVID-19 to the worse clinical course and higher mortality. COVID-19 is a systemic disease involving not only the lungs but also the abdominal organs. Thus, the clinical symptoms might be variable.

## Figures and Tables

**Table 1 jcm-11-01821-t001:** Baseline characteristics of 2184 patients hospitalized due to COVID-19, divided into four study groups.

Variables	No Dyspnea and No Abdominal Pain *n* = 1166; (Group A)	Dyspnea *n* = 871; (Group B)	Abdominal Pain *n* = 97; (Group C)	Dyspnea and Abdominal Pain *n* = 50; (Group D)	*p*-Value
Sex:					0.0001
- women	629 (54.0)	388 (44.6)	56 (57.7)	29 (58.0)	
- men	537 (46.0)	483 (55.4)	41 (42.3)	21 (42.0)	
Age (years old)	63.0 (41.0–73.0) ^1^	65.0 (52.0–74.0) ^1,2^	63.0 (38.0–70.0) ^2^	64.0 (40.0–75.0)	0.001

Values for continuous variables were showed as or median (Q25–Q75), and values for categorical variables were presented as number of observation (percent) ^1^: A vs. B, *p* = 0.002; ^2^: B vs. C, *p* = 0.061 tendency to statistical significance.

**Table 2 jcm-11-01821-t002:** Comparison of the clinical and laboratory parameters in four tested groups of patients with COVID-19 at the time of admission to hospital and during hospitalization.

Variables	Control Group *n* = 1166; (Group A)	Dyspnoea *n* = 871; (Group B)	Abdominal Pain *n* = 97; (Group C)	Dyspnoea and Abdominal Pain *n* = 50; (Group D)	*p*-Value
Parameters Obtained at the Admission to the Hospital					
Saturation (SaO_2_) without oxygen therapy (%)	96.0 (94.0–98.0) ^1,2^	90.0 (85.0–95.0) ^1,3^	97.0 (95.0–98.0) ^3,4^	92.4 ± 4.5 ^2,4^	<0.0001
PaO_2_ [mmHg]	61.5 ± 31.4	54.0 (35.5–73.5)	42.0 (26.0–50.0)	49.8 ± 26.4	0.045
Pain in the chest	50 (4.3)	101 (11.6)	3 (3.1)	9 (18.0)	<0.0001
Cough	171 (14.7)	434 (49.8)	15 (15.5)	28 (56.0)	<0.0001
Standstill heard over the lung fields	109 (9.4)	233 (26.8)	10 (10.3)	15 (30.0)	<0.0001
Stressed sounds heard over the lung fields	108 (9.3)	189 (21.7)	10 (10.3)	12 (24.0)	<0.0001
Whistling sounds heard over the lung fields	63 (5.4)	147 (16.9)	4 (4.1)	5 (10.0)	<0.0001
Diarrhea	35 (3.0)	58 (6.7)	21 (21.7)	13 (26.0)	<0.0001
Vomiting	35 (3.0)	21 (2.4)	30 (30.9)	12 (24.0)	<0.0001
Respiratory support:					<0.0001
- oxygen mustache cannula	127 (10.9)	277 (31.9)	8 (8.3)	16 (32.0)	
- face mask	34 (2.9)	104 (12.0)	1 (1.0)	5 (10.0)	
- Venturi mask	3 (0.3)	12 (1.4)	2 (2.1)	0 (0.0)	
- passive oxygen therapy	33 (2.8)	176 (20.3)	2 (2.1)	5 (10.0)	
- HFNC	1 (0.1)	9 (1.0)	0 (0.0)	0 (0.0)	
- BiPAP/CPAP	1 (0.1)	8 (0.9)	0 (0.0)	0 (0.0)	
- respirator therapy	72 (6.2)	11 (1.3)	0 (0.0)	0 (0.0)	
ASPAT (U/L)	32.0 (21.0–55.0) ^5^	43.0 (29.0–69.0) ^5,6^	33.0 (22.0–56.0) ^6^	42.5 (26.0–59.0)	<0.0001
ALAT (U/L)	25.0 (16.0–45.0) ^7^	35.0 (21.0–58.0) ^7^	26.0 (17.0–47.0)	32.0 (20.0–46.0)	<0.0001
Total bilirubin (mg/dL)	0.6 (0.5–0.9)	0.6 (0.5–0.8)	0.6 (0.5–1.0)	0.6 (0.5–0.9)	0.365
Amylase in blood (U/L)	50.5 (31.0–75.0)	50.0 (32.0–72.0)	50.5 (31.0–63.0)	55.0 (42.0–60.0)	0.834
Lipase (U/L)	29.0 (16.0–63.0)	31.0 (18.0–56.0)	30.0 (15.0–63.0)	41.0 (22.0–57.0)	0.742
CRP (mg/L)	30.7 (5.2–89.8) ^8,9^	74.4 (32.7–144.3) ^8,10^	40.7 (11.3–100.8) ^10^	71.2 (31.4–128.15) ^9^	<0.0001
Procalcitonin (ng/mL)	0.09 (0.04–0.31)	0.09 (0.04–0.30)	0.07 (0.04–0.22)	0.07 (0.04–0.26)	0.351
Parameters obtained during hospitalization					
Deterioration of health—The need for maximum aggressive oxygen therapy	329 (28.2)	371 (42.6)	26 (26.8)	17 (34.0)	<0.0001
The most aggressive respiratory support during hospitalization:					<0.0001
- HFNC	32 (2.8)	92 (10.6)	3 (3.1)	4 (8.0)	
- BiPAP/CPAP	9 (0.8)	32 (3.7)	0 (0.0)	1 (2.0)	
- respirator therapy	112 (9.6)	96 (11.0)	3 (3.1)	1 (2.0)	
Whistling/rattling sounds	274 (23.5)	441 (50.7)	20 (20.6)	28 (56.0)	<0.0001
Pneumonia	397 (34.1)	599 (68.8)	32 (33.0)	33 (66.0)	<0.0001
Hypovolemic shock	19 (1.6)	12 (1.4)	4 (4.1)	0 (0.0)	0.199
Cardiac shock	18 (1.5)	13 (1.5)	1 (1.0)	0 (0.0)	0.643
Septic shock	80 (6.9)	57 (6.5)	4 (4.1)	0 (0.0)	0.046
Digastive tract hemorrhage:					0.500
- upper part	17 (1.5)	12 (1.4)	3 (3.1)	0 (0.0)	
- lower part	4 (0.3)	3 (0.3)	1 (1.0)	1 (2.0)	
Respiratory hemorrhage	14 (1.2)	17 (2.0)	1 (1.0)	2 (4.0)	0.330
The need for intubation	111 (9.5)	100 (11.5)	3 (3.1)	1 (2.0)	0.003
Transfer to the Intensive Care Unit	106 (9.1)	101 (11.6)	5 (5.2)	2 (4.0)	0.031
Ventilation mode:					0.507
- A/C	65 (65.0)	39 (54.9)	1 (50.0)	0 (0.0)	
- CMV	19 (19.0)	12 (16.9)	1 (50.0)	0 (0.0)	
- SIMS	16 (16.0)	20 (28.2)	0 (0.0)	0 (0.0)	
Hospitalization:					0.003
- Discharge home	741 (63.6)	481 (55.2)	65 (67.0)	29 (58.0)	
- Transfered to another hospital for specialist treatment or deterioration of health	129 (11.1)	128 (14.7)	14 (14.4)	9 (18.0)	
- Transfered to another hospital for rehabilitation	143 (12.3)	104 (11.9)	8 (8.3)	7 (14.0)	
- Patient’s death	153 (13.1)	158 (18.1)	10 (10.3)	5 (10.0)	
ASPAT (U/L)	29.0 (20.0–48.0) ^11^	35.0 (24.0–57.0) ^11^	28.0 (21.0–45.0)	34.0 (24.0–58.0)	<0.0001
ALAT (U/L)	28.0 (17.0–53.0) ^12^	41.0 (22.0–75.0) ^12,13^	28.0 (15.0–48.0) ^13^	39.0 (20.0–61.0)	<0.0001
Total bilirubin (mg/dL)	0.6 (0.4–0.8)	0.6 (0.5–0.8)	0.6 (0.4–0.9)	0.6 (0.4–0.9)	0.930
Amylase in blood (U/L)	49.5 (30.0–71.0)	50.0 (34.0–69.0)	45.5 (28.0–65.0)	52.7 ± 15.9	0.805
Lipase (U/L)	30.0 (16.0–65.0)	34.0 (19.0–58.0)	33.0 (15.0–63.0)	41.0 (21.0–54.0)	0.876
CRP (mg/L)	15.6 (3.9–65.6) ^14^	19.6 (5.0–94.4) ^14^	20.7 (6.6–54.0)	28.9 (7.4–94.7)	0.006
Procalcitonin (ng/mL)	0.06 (0.03–0.30)	0.05 (0.03–0.2)	0.05 (0.03–0.13)	0.06 (0.03–0.14)	0.292

HFNC, high flow nasal cannula; BiPAP/CPAP, bilevel positive airway pressure/continuous positive airway pressure; A/C, assist/control; CMV, continuous mandatory ventilation; SIMV, synchronized intermittent mandatory ventilation; ASPAT, aspartate transaminase; ALAT, alanine aminotransferase; CRP, C-reactive protein. Values for continuous variables were showed as mean ± SD and for variables without normal distribution as median (Q25-Q75), and values for categorical variables were presented as number of observation (percent). ^1^: A vs. B, *p* < 0.0001, ^2^: A vs. D, *p* = 0.0004; ^3^: B vs. C, *p* < 0.0001; ^4^: C vs. D, *p* < 0.001; ^5^: A vs. B, *p* < 0.0001; ^6^: B vs. C, *p* = 0.009; ^7^: A vs. B, *p* < 0.0001; ^8^: A vs. B, *p* < 0.0001; ^9^: A vs. D, *p* < 0.001; ^10^: B vs. C, *p* < 0.001; ^11^: A vs. B, *p* < 0.0001; ^12^: A vs. B, *p* < 0.0001; ^13^: B vs. C, *p* = 0.003; ^14^: A vs. B, *p* = 0.013.

**Table 3 jcm-11-01821-t003:** Logistic regression analysis of membership in selected groups of COVID-19 patients as predictors of patient’s death.

Groups	OR (± 95% CI)	*p*-Value
gr. B vs. gr. A (for gr. B)	1.47 (1.15–1.87)	0.002
gr. C vs. gr. A (for gr. C)	0.87 (0.62–1.22)	0.428
gr. B vs. gr. C (for gr. C)	0.52 (0.26–1.02)	0.057

Qualifying the patient to group B, but not to group C, significantly influenced the prevalence of patient’s death.

**Table 4 jcm-11-01821-t004:** Univariable and multivariable logistic regression analysis of selected variables as predictors of death in group of patients with dyspnea (Group B; *n* = 871).

Risk Parameters	Univariable		Multivariable	
	OR (±95% CI)	*p*-Value	OR (±95% CI)	*p*-Value
Parameters at the Time of Admission to the Hospital:				
Sex (for men)	1.44 (1.01–2.05)	0.045	1.55 (1.05–2.29)	0.027
Age (for ≥65 years old)	4.13 (2.78–6.15)	<0.0001	3.35 (2.20–5.11)	<0.0001
Pain in the chest	1.13 (0.67–1.90)	0.644		
Cough	0.43 (0.29–0.61)	<0.0001	0.52 (0.35–0.77)	0.001
Standstill heard over the lung fields	2.51 (1.75–3.59)	<0.0001	2.10 (1.41–3.13)	<0.001
Stressed sounds heard over the lung fields	1.77 (1.20–2.61)	0.003	1.35 (0.87–2.10)	0.174
Whistling sounds heard over the lung fields	2.00 (1.32–3.02)	<0.001	1.49 (0.95–2.33)	0.082
Diarrhea	0.93 (0.46–1.90)	0.854		
Vomiting	1.06 (0.35–3.22)	0.913		
Respiratory suport:				
Oxygen mustache	0.48 (0.31–0.73)	<0.001	0.65 (0.40–1.05)	0.081
Face mask	1.24 (0.74–2.05)	0.403		
Venturi mask	3.28 (1.02–10.51)	0.044	3.37 (0.99–11.47)	0.051
Passive oxygen therapy	2.41 (1.64–3.54)	<0.0001	1.91 (1.21–3.01)	0.005
HFNC	5.77 (1.53–21.80)	0.009	6.62 (1.56–27.96)	0.010
BiPAP/CPAP	1.18 (0.92–1.50)	0.171		
Respirator therapy	1.70 (0.44–6.49)	0.437		
Laboratory parameters at admission:				
ASPAT (for: >40 U/L)	1.44 (0.98–2.13)	0.059	1.49 (0.59–3.79)	0.395
ALAT (for: >40 U/L)	0.65 (0.44–0.96)	0.031	0.67 (0.29–1.68)	0.419
Total bilirubin (for: >1.1 mg/dL)	2.80 (1.57–4.98)	<0.0001	1.93 (0.71–5.23)	0.194
Amylase in blood (for: >160 U/L)	19.78 (2.38–163.90)	0.005	10.47 (1.05–104.63)	0.044
Lipase (for >150 U/L)	1.20 (0.34–4.09)	0.772		
CRP (for: >5 mg/L)	2.22 (0.78–6.32)	0.134		
Procalcitonin (for: > 0.1 ng/mL)	5.23 (3.41–8.03)	<0.0001	6.13 (2.49–15.06)	<0.0001
Parameters during hospitalization:				
The most aggressive respiratory support during hospitalization:				
HFNC	3.45 (2.17–5.48)	<0.0001	16.38 (6.71–39.93)	<0.0001
BiPAP/CPAP	6.41 (3.11–13.20)	<0.0001	46.04 (15.55–136.23)	<0.0001
Respirator therapy	6.70 (4.27–10.51)	<0.0001	1.41 (0.10–19.31)	0.792
Whistling/rattling sounds	0.34 (0.23–0.49)	<0.0001	2.63 (1.14–6.07)	0.022
Pneumonia	2.19 (1.42–3.36)	0.0003	1.01 (0.68–1.48)	0.963
Hypovolemic shock	6.56 (2.1–20.99)	0.001	2.54 (0.37–17.26)	0.338
Cardiac shock	26.50 (5.82–121.54)	<0.0001	14.34 (2.32–88.46)	0.004
Septic shock	26.21 (13.16–52.17)	<0.0001	10.45 (2.54–43.01)	0.001
Digastive tract hemorrhage:				
- upper part	3.29 (1.03–10.54)	0.044	2.51 (0.57–11.07)	0.221
- lower part	2.26 (0.20–25.21)	0.505		
Respiratory hemorrhage	4.17 (1.58–11.00)	0.004	1.59 (0.42–5.99)	0.491
Transferred to the Intensive Care Unit	5.17 (3.32–8.02)	<0.0001	5.06 (3.01–8.28)	<0.001
The need for intubation	7.53 (4.82–11.74)	<0.0001	46.95 (5.09–432.87)	<0.001
Ventilation mode:				
- A/C	0.96 (0.36–2.55)	0.933		
- CMV	2.06 (0.49–8.60)	0.314		
- SIMV	0.67 (0.22–1.94)	0.449		
Laboratory parameters during hospitalization:				
ASPAT (for: >40 U/L)	2.89 (1.96–4.27)	<0.0001	2.27 (1.26–4.08)	0.006
ALAT (for: >40 U/L)	0.85 (0.58–1.23)	0.385		
Total bilirubin (for: >1.1 mg/dL)	5.33 (3.08–9.22)	<0.0001	4.99 (1.91–13.08)	0.001
Amylase in blood (for: >160 U/L)	1 ref.	-		
Lipase (for >150 U/L)	0.88 (0.22–3.41)	0.852		
CRP (for: >5 mg/L)	66.67 (9.24–481.12)	<0.0001	23.84 (3.06–185.30)	0.002
Procalcitonin (for: > 0.1 ng/mL)	30.64 (17.66–53.18)	<0.0001	22.56 (11.86–42.91)	<0.0001

HFNC, high flow nasal cannula; BiPAP/CPAP, bilevel positive airway pressure/continuous positive airway pressure; A/C, assist/control; CMV, continuous mandatory ventilation; SIMV, synchronized intermittent mandatory ventilation; ASPAT, aspartate transaminase; ALAT, alanine aminotransferase; CRP, C-reactive protein; OR, odds ratio; ±95%CI, 95% confidence intervals.

**Table 5 jcm-11-01821-t005:** Univariable and multivariable logistic regression analysis of selected variables as predictors of death in group of patients with abdominal pain (*n* = 97).

Risk Parameters	Univariable		Multivariable	
	OR (±95% CI)	*p*-Value	OR (±95% CI)	*p*-Value
Parameters at the Admission to the Hospital:				
Sex (for men)	0.90 (0.23–3.48)	0.878		
Age (for ≥65 years old)	2.34 (0.60–9.05)	0.213		
Pain in the chest	4.72 (0.38–59.22)	0.223		
Cough	1 ref.	-		
Standstill heard over the lung fields	2.46 (0.43–13.97)	0.300		
Stressed sounds heard over the lung fields	0.96 (0.11–8.79)	0.972		
Whistling sounds heard over the lung fields	3.11 (0.28–34.14)	0.346		
Diarrhea	0.37 (0.04–3.20)	0.362		
Vomiting	0.95 (0.22–4.04)	0.946		
Respiratory suport:				
Oxygen mustache cannula	3.38 (0.56–20.01)	0.175		
Face mask	1 ref.	-		
Venturi mask	9.55 (0.53–172.35)	0.121		
Passive oxygen therapy	1 ref.	-		
HFNC	1 ref.	-		
BiPAP/CPAP	1 ref.	-		
Respiratory therapy	1 ref.	-		
Laboratory parameters at admission:				
ASPAT (for: >40 U/L)	1.83 (0.42–7.91)	0.406		
ALAT (for: >40 U/L)	2.14 (0.51–9.05)	0.290		
Total bilirubin (for: >1.1 mg/dL)	0.86 (0.09–8.01)	0.890		
Amylase in blood (for: >160 U/L)	1 ref.	-		
Lipase (for >150 U/L)	1 ref.	-		
CRP (for: >5 mg/L)	1 ref.	-		
Procalcitonin (for: > 0.1 ng/mL)	8.00 (1.41–45.21)	0.016	8.00 (1.41–45.21)	0.016
Parameters during hospitalization:				
The most aggressive respiratory support during hospitalization:				
Whistling/rattling sounds	1.76 (0.40–7.68)	0.443		
HFNC	1 ref.	-		
BiPAP/CPAP	1 ref.	-		
Respirator therapy	21.50 (1.69–272.56)	0.016	0.085 (0.00–15.35)	0.346
Pneumonia	5.78 (1.35–24.63)	0.016	6.20 (0.92–41.74)	0.057
Hypovolemic shock	10.62 (1.27–88.22)	0.027	1.84 (0.02–138.23)	0.779
Cardiac shock	1 ref.	-		
Septic shock	36.85 (3.27–415.19)	0.003	1.31 (0.04–47.73)	0.882
Digastive tract hemorrhage:- upper part- lower part	4.72 (0.37–59.22)1 ref.	0.223-		
Respiratory hemorrhage	1 ref.	-		
Transferred to the Intensive Care Unit	18.21 (2.52–131.04)	0.003	10.35 (0.13–848.87)	0.292
Laboratory parameters during hospitalization:				
ASPAT (for: >40 U/L)	0.65 (0.11–3.60)	0.620		
ALAT (for: >40 U/L)	0.62 (0.11–3.33)	0.574		
Total bilirubin (for: >1.1 mg/dL)	2.26 (0.38–13.32)	0.360		
Amylase in blood (for: >160 U/L)	1 ref.	-		
Lipase (for >150 U/L)	1 ref.	-		
CRP (for: >5 mg/L)	1 ref.	-		
Procalcitonin (for: > 0.1 ng/mL)	16.62 (2.78–99.32)	0.002	16.62 (2.78–99.32)	0.002

HFNC, high flow nasal cannula; BiPAP/CPAP, bilevel positive airway pressure/continuous positive airway pressure; ASPAT, aspartate transaminase; ALAT, alanine aminotransferase; CRP, C-reactive protein; OR, odds ratio; ±95% CI, 95% confidence intervals.

## Data Availability

The datasets used and/or analyzed during the current study are available from the corresponding author upon reasonable request.

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
