# Peer review of "Assessment of Gastrointestinal Symptoms and Dyspnea in Patients Hospitalized due to COVID-19: Contribution to Clinical Course and Mortality"

_jcm, 2022, doi:10.3390/jcm11071821_

Round 1

Reviewer 1 Report

  • Use Oxford comma

  • “Study group consists of 2,184 COVID-19 patients. There were 1,102 (50.5%) women 127

and 1,082 (49.5%) men, with mean age 60.1 + 18.8 years old. The patients were divided 128

into 4 groups according to presented symptoms at the time of admission to the hospital. 129

Group A: patients without dyspnea and AP, n=1,166 (53.4%), (study control population); 130

group B: patients with dyspnea, n=871 (39.9%); group C: patients with AP, n=97 (4.4%) 131

and group D: patients with dyspnea and AP, n=50 (2.3%).”

“Groups were differ in sex (p=0.0001) and age (p=0.001) parameters (Table 1), 137

therefore subsequent data were analyzed separately according sex. The “age” variable 138

was not significant in selected “women” and “men” groups in post-hoc tests (p>0.05).”

Do not report the number of included patients and their baseline characteristics in the Material section, insert in the Results section

  • “Values for continuous variables were showed as mean + SD and for variables without normal distribution as median (Q25-Q75), and values for categorical variables were presented as number of observation (percent).

1: A vs B, p<0.0001, 2: A vs D, p=0.0004; 3: B vs C, p<0.0001; 4: C vs D, p<0.001; 5: A vs

B, p<0.0001; 6: B vs C, p=0.009; 7: A vs B, p<0.0001; 8: A vs B, p<0.0001; 9: A vs D, p<0.001; 10: B vs C, p<0.001; 11: A vs B, p<0.0001; 12: A vs B, p<0.0001; 13: B vs C, p=0.003; 14: A vs B, p=0.013”

Is this a note to Table 2 or a part of results?

  • “of death in group. of patients with dyspnea”

Delete the point and uniform the character

Author Response

Reviewer 1#

We would like to thank the reviewer very much for the constructive criticism and the comments. Thank you very much. Regarding the concrete suggestions we answered them point-by-point as follows:    

  • “Use Oxford comma”

Ad.1 Dear reviewer, the manuscript was finally prepared and corrected by American Journal Experts (Certificate attached), so we trusted them and sent the manuscript as prepared. However we see, they did not use Oxford comma. We corrected it, and introduced Oxford comma within the whole paper according suggestion. Thank you.         

  • “Study group consists of 2,184 COVID-19 patients. There were 1,102 (50.5%) women and 1,082 (49.5%) men, with mean age 60.1 + 18.8 years old. The patients were divided into 4 groups according to presented symptoms at the time of admission to the hospital. Group A: patients without dyspnea and AP, n=1,166 (53.4%), (study control population); group B: patients with dyspnea, n=871 (39.9%); group C: patients with AP, n=97 (4.4%) and group D: patients with dyspnea and AP, n=50 (2.3%).”

“Groups were differ in sex (p=0.0001) and age (p=0.001) parameters (Table 1), therefore subsequent data were analyzed separately according sex. The “age” variable was not significant in selected “women” and “men” groups in post-hoc tests (p>0.05).”

Do not report the number of included patients and their baseline characteristics in the Material section, insert in the Results section

Ad. 2 We modified and moved the paragraphs from the Materials and Methods section to Results section. Thank you.

  • “Values for continuous variables were showed as mean + SD and for variables without normal distribution as median (Q25-Q75), and values for categorical variables were presented as number of observation (percent). 1: A vs B, p<0.0001, 2: A vs D, p=0.0004; 3: B vs C, p<0.0001; 4: C vs D, p<0.001; 5: A vs B, p<0.0001; 6: B vs C, p=0.009; 7: A vs B, p<0.0001; 8: A vs B, p<0.0001; 9: A vs D, p<0.001; 10: B vs C, p<0.001; 11: A vs B, p<0.0001; 12: A vs B, p<0.0001; 13: B vs C, p=0.003; 14: A vs B, p=0.013”

Is this a note to Table 2 or a part of results?

Ad. 3 The information above is the note to Table 2, not a part of Results section. In the original version of the manuscript it was placed correctly. After formatting the manuscript by the editor it was changed. We corrected it again. 

  • “of death in group. of patients with dyspnea”

Delete the point and uniform the character

 Ad. 4 In the original version of the manuscript it was written correctly. After formatting the manuscript by the editor it was changed. We corrected it again. 

Dear reviewer,

thank you very much for this detailed, accurate and extremely helpful review.

Thank you.

Reviewer 2 Report

The authors analyze an important and up to date topic in today’s medicine: COVID and its symptoms. An attempt is made to analyze covid as a systemic disease based on its symptoms, but the authors fail a crucial point in the manuscript: all the patients presenting with abdominal pain as a symptom, associated with dyspnea or not are considered to have AP due to the COVID infection.

The authors fail to properly explain the difference between abdominal pain in COVID and abdominal pain in any other pathology (acute in nature) and therefore the premises of the study is wrong.

Author Response

Reviewer 2#

We would like to thank the reviewer very much for the statements, that “The authors analyze an important and up to date topic in today’s medicine: COVID and its symptoms.” This opinion is incredible important for us. Thank you very much. Regarding the concrete suggestions we answered them point-by-point as follows:

  • The authors analyze an important and up to date topic in today’s medicine: COVID and its symptoms. An attempt is made to analyze covid as a systemic disease based on its symptoms, but the authors fail a crucial point in the manuscript: all the patients presenting with abdominal pain as a symptom, associated with dyspnea or not are considered to have AP due to the COVID infection.

  • The authors fail to properly explain the difference between abdominal pain in COVID and abdominal pain in any other pathology (acute in nature) and therefore the premises of the study is wrong.

Ad. 1 Dear reviewer, at the very beginning we would like to apologize for the not clearly formed and presented in the study the exclusion criteria of the analyzed patients. The lack of them makes some confusions and misunderstanding. 

It was estimated, that on January 19th, 2020, in Washington (USA), the first COVID-19-positive patient manifested, in addition to respiratory signs, abdominal symptoms such as abdominal pain, nausea and vomiting [1]. What is more, some authors noticed, that several individuals with COVID-19, who presented to the emergency department had only abdominal pain without any typical respiratory signs characteristic of this infection [2]. On basis of this information, we performed a retrospective analysis of individuals, who suffered with COVID-19 to determine whether, and how many patients infected by SARS-CoV-2 manifested abdominal pain. We checked, whether the patients with COVID-19 disease, who presented with abdominal pain, had significantly elevated specific markers for digestive system involvement. We assessed the correlation between respiratory symptoms and abdominal pain to establish how often they occur together and separately. We assessed how abdominal pain may influence the clinical course and mortality of patients with COVID-19. The patients, who presented abdominal pain as the symptom of the acute abdominal diseases, which may produce abdominal pain like appendicitis, cholecystitis, diverticulitis, incarcerated or strangulated hernias, occlusion of mesenteric artery or aortic aneurysm and were also infected by SARS-CoV-2 were excluded from the study. They were not admitted to the Temporary COVID-19 University Hospital, but they were transported to the Department of Surgery dedicated to SARS-CoV-2 infected patients for potential surgical intervention. We included in the Materials and Methods section “the exclusion criteria” as follows: “The patients infected by SARS-CoV-2 in whom we confirmed acute abdominal disease like appendicitis, cholecystitis, diverticulitis, incarcerated and strangulated hernia, occlusion of mesenteric artery or aortic aneurysm were excluded from the study. They were transported to the Department of Surgery for potential surgical intervention. They were not admitted to the temporary COVID-19 hospital.” We attached the flow diagram as the supplementary material to present the patients’ exclusion criteria. 

Dear reviewer, thank you very much for this accurate and helpful review. Thank you.

References

  1. Holshue, M.L.; DeBolt, C.; Lindquist, S.; Lofy, K.H.; Wiesman, J.; Bruce, H.; Spitters, C.; Ericson, K.; Wilkerson, S.; Tural, A.; et al. First case of 2019 novel coronavirus in the United States. Engl. J. Med. 2020, 382, 929–936; DOI:10.1056/NEJMoa2001191.
  2. Boraschi, P.; Giugliano, L.; Mercogliano, G.; Donati, F.; Romano, S.; Neri, E. Abdominal and gastrointestinal manifestations in COVID-19 patients: Is imaging useful? World J. Gastroenterol. 2021, 27, 4143–4159; DOI:10.3748/wjg.v27.i26.4143.
